# Investigation of the Therapeutic Potential of New Antidiabetic Compounds Using Islet-on-a-Chip Microfluidic Model

**DOI:** 10.3390/bios12050302

**Published:** 2022-05-05

**Authors:** Patrycja Sokolowska, Elzbieta Jastrzebska, Agnieszka Dobrzyn, Zbigniew Brzozka

**Affiliations:** 1Chair of Medical Biotechnology, Faculty of Chemistry, Warsaw University of Technology, 00-661 Warsaw, Poland; elzbieta.jastrzebska@pw.edu.pl (E.J.); zbigniew.brzozka@pw.edu.pl (Z.B.); 2Laboratory of Cell Signaling and Metabolic Disorders, Nencki Institute of Experimental Biology, Polish Academy of Sciences, 02-093 Warsaw, Poland; a.dobrzyn@nencki.edu.pl

**Keywords:** islet-on-a-chip model, therapeutic agent, 5-PAHSA, 9-PAHSA, fatty acid, glucose stimulated insulin secretion, GSIS, Lab-on-a-chip model, glucagon secretion

## Abstract

Nowadays, diabetes mellitus is one of the most common chronic diseases in the world. Current research on the treatment of diabetes combines many fields of science, such as biotechnology, transplantology or engineering. Therefore, it is necessary to develop new therapeutic strategies and preventive methods. A newly discovered class of lipids—Palmitic Acid Hydroxy Stearic Acid (PAHSA) has recently been proposed as an agent with potential therapeutic properties. In this research, we used an islet-on-a-chip microfluidic 3D model of pancreatic islets (pseudoislets) to study two isomers of PAHSA: 5-PAHSA and 9-PAHSA as potential regulators of proliferation, viability, insulin and glucagon expression, and glucose-stimulated insulin and glucagon secretion. Due to the use of the Lab-on-a-chip systems and flow conditions, we were able to reflect conditions similar to in vivo. In addition, we significantly shortened the time of pseudoislet production, and we were able to carry out cell culture, microscopic analysis and measurements using a multi-well plate reader at the same time on one device. In this report we showed that under microfluidic conditions PAHSA, especially 5-PAHSA, has a positive effect on pseudoislet proliferation, increase in cell number and mass, and glucose-stimulated insulin secretion, which may qualify it as a compound with potential therapeutic properties.

## 1. Introduction

The World Health Organization (WHO) predicts that by 2030 diabetes mellitus (DM) will be the seventh leading cause of death in the world [1]. This disease is closely related to the pancreas, and more specifically to the pancreatic islet, which is a cluster of endocrine cells that synthesize and release hormones. Five types of cells build the pancreatic islet, but two of them are particularly important in the course of diabetes—glucagon secreting α-cells and insulin secreting β-cells [2]. Only the proper functioning of the pancreas and the proper secretion of hormones ensures the body’s homeostasis. The causes of diabetes mellitus type 2 are not fully understood. The risk factors include, among others, genetic and environmental factors such as inadequate diet, lack of physical activity, and stress [3]. For this reason, the number of studies of new diagnostic and therapeutic strategies have recently increased. So far, significant progress has been made, including the development of pancreatic islet models imitating in vivo conditions, especially with the use of Lab-on-chip systems [4,5,6,7] drug screening [8,9] or new treatment strategies [10,11,12,13,14]. However, there is still a need to find effective and safe therapeutic agents. Recently discovered class of lipids named FAHFA (Fatty Acyl esters of Hydroxy Fatty Acids) seems to be a promising compound in the treatment of diabetes and inflammatory diseases. They are endogenous lipids with properties that increase glucose tolerance, increase the secretion of insulin and GLP-1 (glucagon-like peptide 1), and reduce the inflammatory response [15]. These lipids occur naturally in the body in adipose tissue, kidneys, liver, serum, breast milk, or meconium. They have beneficial effects on target cells and tissues in immune cells, adipocytes, intestines, or pancreatic islets. Both mice and human FAHFA levels fluctuate from 0.5 to 500 nmol/L in serum/plasma and up to 200 pmol/g in tissue [16].

There are several dozen of FAHFA families that are characterized by a different composition of fatty acids and hydroxy fatty acids [15]. PAHSA (Palmitic Acid Hydroxy Stearic Acid), which consist of palmitic acid (PA) esterified to the hydroxyl group of hydroxystearic acid (HSA), is one of the FAHFA families. Regioisomer (e.g., 5-PAHSA, 9-PAHSA) is determined by the location of the branched carbon [17]. PAHSA has been shown to have a positive effect on organs/tissues such as: adipose tissue [18], pancreas [19], gut [20], liver [21], muscles [22]. The effect of PAHSA on the pancreas and its potential use as a therapeutic agent in the treatment of diabetes seems to be of particular interest. It was showed that PAHSA has the beneficial effect on pancreatic β-cells proliferation, as well as increased insulin and glucagon-like peptide 1 (GLP-1) secretion, and thus a possible anti-diabetic effect [19]. Interestingly, PAHSA levels in the serum and adipose tissue of insulin-resistant humans and high-fat diet fed (HFD) mice have been found to be reduced. It was showed that chronic administration of 5-PAHSA and 9-PAHSA to HFD mice increases PAHSA levels by ~1.4 to 3-fold, resulting in improved insulin sensitivity and glucose tolerance [22]. Moreover, there are indications that the administration of solutions containing PAHSA may attenuate cytokine-induced apoptotic and necrotic β-cell death and increase β-cell viability [23]. The mechanism of PAHSA action is not yet fully understood, but it suggests the activation of the GPR40 receptor, which is a member of the family of G-protein coupled free fatty acid receptors and is activated by saturated and unsaturated carboxylic acids. In addition, it stimulates the influx of Ca2+ ions and increases the secretion of insulin and glucagon-like peptide 1 (GLP-1). However, some studies suggest that high glucose levels in the blood may reduce the effect of 5-PAHSA by inhibiting the AMPK signaling pathway and promoting nuclear factor kappa-B (NF-κB) mediated inflammation [24]. There are also studies that question antidiabetic potential of PAHSA [18]. Therefore, it is very important to resolved contentious data on the role of PAHSA in the regulation of pancreatic islet function. In our study, we assessed the possibility of using the previously developed pseudoislet model in Lab-on-a-chip system as a universal model for the rapid testing of new therapeutic agents [5,25]. Simultaneously, we investigated the influence of two PASHA isomers (5-PAHSA, 9-PAHSA) on the functionality and insulin/glucagon secretion from a three-dimensional pseudoislet model in Lab-on-a-chip system. The positive effect of PAHSA isomers on pancreatic islet cell proliferation, aggregation process, and increase their mass was confirmed. The ability of 5-PAHSA and 9-PAHSA to increase insulin and glucagon secretion, both after stimulation with low and high glucose concentrations and without them, was also noted. Thus, we believe that PAHSA (especially 5-PASHA) can be considered as a potential therapeutic agent in the treatment of diabetes. In addition, it was confirmed that due to the developed geometry, our system can be a good candidate as a universal tool for research newly discovered therapeutic agents. Thanks to the geometry of the system and the selection of appropriate culture parameters, it is possible to obtain a stable culture of pseudoislets, reflect of in vivo conditions, shorten the time of the research and analysis in real time on one device. To the best of our knowledge, this is the first report that determinates the direct effect of treatment with 5-PAHSA and 9-PAHSA on a pseudoislet model in a microfluidic system.

## 2. Materials and Methods

### 2.1. Three-Dimensional Islet-on-a-Chip Model

The three-dimensional model of the pseudoislet was developed by using a microfluidic system composed of two poly(dimethylsiloxane) (PDMS, Sylgard 184, Dow Corning, Wiesbaden, Germany) layers, which were connected to each other by oxygen plasma treatment (Preen II-973, Plasmatic System, Inc.). The geometry, materials, and fabrication methods of the microfluidic system used in this study were described in detail in our earlier work [5]. In short, the geometry of the microfluidic system contained two main microchamber (9000 µm length, 6000 µm width, 200 µm height), each of them was equipped with 15 round microtraps (280 µm diameter, 200 µm height) which limit cell growth surfaces and thus support aggregation. Each of the microtraps was constructed of seven circular micropillar (145 µm × 145 µm × 200 µm), arranged in a semicircle at 20 µm intervals with 160 µm wide open inlet space. The islet-on-a-chip model was developed by using the above-described microsystem and two pancreatic islet cell lines: α-cells (α-TC1-6) were purchased from the American Type Culture Collection (catalog no. CRL-2934) and rat INS-1E insulinoma β-cell line was a gift from Dr. Pierre Maechler (University of Geneva, Geneva, Switzerland). The culture conditions have been described in detail previously [5]. As in previous studies, in the Lab-on-a-chip system, cells were co-cultured in the ratio 1: 2 (α-TC1-6 cell suspension (density of 2 × 10^6^ cells/mL) and INS-1E cell suspension (2 × 10^6^ cells/mL)). In each chamber of the islet-on-a-chip system, 15 spherical aggregates were obtained, which in terms of the composition and localization of α and ß cells correspond to the rodent pancreatic islet in vivo were used for all further studies. The microchamber geometry, complete system, and the obtained cell structures after 24 h of culture are shown in Figure 1.

### 2.2. The Influence of PAHSA on Cell Proliferation

Our goal was to test a wide range of concentrations (5 µM, 20 µM, 40 µM, 60 µM, 80 µM, 100 µM) of two regioisomers of palmitic acid hydroxy stearic acid. We used 5-(palmitoyloxy) octadecanoic acid (5-PAHSA, Cayman Chemicals) and 9-[(1-oxohexadecyl)oxy]-octadecanoic acid (9-PAHSA, Cayman Chemicals) in all studies. After obtaining a spherical pseudoislet model (24 h after the introduction of α- and β-cells cells in ratio of 1:2 into the microfluidic system), solutions of 5-PAHSA or 9-PAHSA at concentrations of 5 µM, 20 µM, 40 µM, 60 µM, 80 µM, 100 µM were introduced into the system. All reagents were introduced into the microfluidic system using a peristaltic pump at a flow rate of 10 μL/min over 3 min. These parameters were selected on the basis of simulation (using Microelectromechanical Systems (MEMS) simulation module of COM- SOL Multiphysics software) and confirmed experimentally. The results were presented in our previous publication [5]. The proliferation rate was measured daily for 5 days of cell culture after 24 h incubation of pseudoislets with above mentioned concentrations of 5- and 9-PAHSA. Each day of the culture pseudoislets were incubate for 50 min with 10% vol AlamarBlue (Abcam) prepared in the culture medium (mixture of INS-1E and α-TC1-6 culture media in a 1:1 ratio). Fluorescence measurements were carried out on a chip using multi-well plate reader (Tecan Infinite 200 Pro) at an excitation wavelength of 552 nm and emission wavelength of 583 nm. Fresh solutions of 5-PAHSA or 9-PAHSA were administered daily. Moreover, every day, after incubation with PAHSA solutions, we conducted microscopic observations of aggregation and changes in the diameter of pseudoislet. Observations were made using an inverted microscope coupled with a CCD camera (Olympus IX70). The microscopic photos were analyzed using CellSens Dimension image analysis software (Olympus).

### 2.3. Immunostaining

To confirm the proper morphology of cell aggregates and to analyze the fluorescence intensity profile after incubation with 5-PAHSA and 9-PAHSA solutions, immunofluorescence staining with primary and secondary antibodies was performed. After the membrane permeabilization and cells fixation steps, primary antibodies for insulin (Cell Signaling Technology) and glucagon (Abcam) in a 1:200 dilution were introduced into the microfluidic system. As a secondary antibodies anti-mouse Alexa Fluor 594 (Thermo Fisher) and anti-rabbit Alexa Fluor 488 (Thermo Fisher) in a 1:200 dilution was used. DNA was stained with Hoechst (Sigma Aldrich). The results were analyzed in two dimensions (2D and 3D views) using a Fluoview FV10i confocal microscope (Olympus) and the Olympus Fluoview Fv10i software.

### 2.4. The Effect of 5- and 9-PAHSA on Glucose Stimulated Insulin Secretion (GSIS)

To examine the therapeutic effect of 5-PAHSA and 9-PAHSA on the pseudoislet model, an islet functionality test was performed after incubation with 5, 20, 40, 60, 80, or 100 µM of 5-PAHSA or 9-PAHSA solutions and stimulation with low (2.75 mM) or high (16.5 mM) glucose concentrations. After obtaining spherical pseudoislets aggregates, the 5-PAHSA or 9-PAHSA solution at the appropriate concentration was introduced (flow rate = 10 μL/min, 3 min) into one chamber of the system, and the culture medium was introduced into the other chamber (as a control). After 48 h of incubation, the medium was removed from the chamber by washing the system with DPBS (Biowest, MS00QC1001) at a flow rate of 10 μL/min, 3 min. Afterward, 2.75 glucose solution (Sigma-Aldrich) prepared in Krebs buffer (135 mM NaCl, 3.6 mM KCl, 5 mM NaHCO_3_, 0.5 mM MgCl_2_ × 6H_2_O, 1.5 mM CaCl_2_ × 2H_2_O, 10 mM HEPES, 0.1% BSA, and 0.5 mM Na_2_PO_4_ × H_2_O) was introduced into all chambers of the microfluidic systems (a flow rate of 10 μL/min, 3 min) and incubated (37 °C, 5% CO_2_) for 1 h. After the incubation step, 2.75 mM glucose was added (10 μL/min, 3 min) into both chambers of one microfluidic system, and Krebs buffer that contained 16.5 mM glucose was added to the other microfluidic system (10 μL/min, 3 min), and incubated for 1 h at 37 °C with 5% CO_2_. Next, the obtained samples were transferred into the Eppendorf tubes and analyzed using the Rat/Mouse Insulin ELISA Kit (Millipore) in a multi-well plate reader (Tecan Infinite 200 Pro), absorbance (450 nm and 590 nm).

### 2.5. The Effect of 5- and 9-PAHSA on Glucagon Secretion

The effect of 5- and 9-PAHSA on glucose secretion was determined from the samples obtained in the previous section (2.4) using the Glucagon Chemiluminescent ELISA Kit (Millipore,) according to the manufacturer’s instructions. Luminescence (~425 nm) was read in a multi-well plate reader (Tecan Infinite 200 Pro).

### 2.6. Statistical Analysis

All the quantitative data were expressed as mean ± standard deviation (SD), based on at least three independent experiments. The statistical analysis was performed using one-way analysis of variance (ANOVA). Values of *p* < 0.05 were considered statistically significant.

## 3. Results and Discussion

### 3.1. The Influence of PAHSA Isomers on Cell Proliferation in 3D Pseudoislet Model

The growing number of diabetes cases and complications related to its development was an impulse to research new therapeutic and prophylactic strategies. Recently, more and more attention has been paid to the influence of FAHFA, especially PAHSA, on improving pancreatic β-cells function. The presence of these substances has been confirmed in the tissues of mammals, but also in nutrients such as fruit, vegetables, eggs and oats [26]. It was concluded that introducing PAHSA-rich foods into the diet or administering this compound orally could be one of the therapeutic strategies for type 2 diabetes. It was shown that oral treatment of nonobese diabetic (NOD) mouse with PAHSA compounds resulted in an increase in the area of β-cells per islet and therefore, an increase in their number [23]. Based on these references, we performed research on the direct impact of a wide range of 5-PAHSA and 9-PAHSA concentrations on the pseudoislet model in flow conditions. In our previous study, due to the use of a microfluidic system with an appropriate geometry, we developed a model that mimics the unique structure of the rodent pancreatic islet (in terms of size, structure of aggregate and distribution of α- and β-cells) was imitated. The model was fully functional, characterized by high viability and proliferation level and an appropriate level of insulin and glucagon secretion under different glucose concentrations [5]. Here we decided to use the above-mentioned model to study the islet cell proliferation, as well as an analysis of islet mass and aggregation process in the following days of PAHSA treatment. The high degree of islet cell aggregates proliferation was detected after incubation with all tested concentrations of 5-PAHSA (Figure 2A). Higher values of proliferation level were obtained in relation to both the control and the stimulation with the same concentrations of 5-PAHSA on the previous day. After 72 h of cell culture, the highest increase in the degree of proliferation (more than twice compared with 24 h of culture) was noticed in the case of stimulation with the 5-PAHSA solution at a concentration of 40 µM and 60 µM (proliferation rate increased by 1.09 and 0.96, respectively). In the case of incubation with 9-PAHSA, in all tested concentrations an approximately two-fold lower increase in the degree of proliferation was observed compared to proliferation after incubation with 5-PAHSA (Figure 2B). The degree of proliferation after 72 h of exposure to 5 µM, 20 µM, 80 µM of 9-PAHSA compared with the first day of stimulation (24 h) increased by 0.15 and 0.3, 0.2, respectively. In the case of higher concentrations 9-PAHSA (especially 60 µM and 100 µM), a decrease in the proliferation level was noted in the following days of culture. In summary, incubations with the 5-PAHSA solutions at all tested concentrations did not have a toxic effect on the 3D structures of pancreatic islets and significantly increased cells proliferation. In contrast, a significant decrease in cell proliferation was noted after incubation with high concentrations of 9-PAHSA. As expected, an increase in diameter and total aggregate mass was noted, which also indicates a high degree of cell proliferation (Figure 2C,D). The diameters of the aggregates, and thus the total mass of β-cells after treatment with 5-PAHSA, increased in the following days of culture. After 72 h of incubation, an increase of about 20–35 µm in relation to the first day was noted. Similarly, in the case of 9-PAHSA, an increase in diameter was also noted in the following days of culture, with a maximum increase of 25 µm. Next days of culture, the area of aggregates increased, which proves that the aggregation process was correct. These results are in line with previous reports on the possible influence of PAHSA isomers on the survival and function of β-cells. The degree of cell proliferation depends on many factors, including age, species, or type of cells. During diabetes, β-cells gradually lose their ability to proliferate, which results in reduced insulin secretion [27]. So far, the effect of PAHSA concentration in the 0–20 µM range on cells proliferation in mice has been tested [21,23]. It was showed that PAHSA reduces apoptotic and necrotic β-cell death and increases their viability. Currently, the proposed mechanism of this phenomenon is the reduction of ER stress and MAPK signaling [23]. Here we showed that PAHSA (and especially the 5-PAHSA isomer) has the direct positive effect on the pancreatic islet cell proliferation (Figure 2).

### 3.2. Study of Pseudoislet Structures and Hormones Fluorescence Intensity

The morphology (i.e., composition and location of cells) and size of the pancreatic islet depend on the species. In rodents, the size of the islet is approximately 200 µm and consists 60% of the β-cells located on the core of the islet, whereas α-cells were distributed peripherally and consisted 25% of all cells in the islet. The remaining 5% of the islet are γ/PP, δ, ε cells [28]. As mentioned before, rodent pancreatic islets are well-defined 3D structures with a specific location and composition of cells. It is very important to maintain the correct morphology of the pseudoislet throughout the culture process. In order to confirm the appropriate aggregate structure and characterize the production of insulin and glucagon after incubation with 5-PAHSA and 9-PAHSA immunofluorescence staining was performed. At this stage of the research, it was decided to choose the three concentration limits: the highest—100, the lowest—5 µM, and the concentration in the middle of the range (60 µM).

The values presented in the graphs were obtained for pseudoislets incubated with the above-mentioned concentrations of 5-PAHSA and 9-PAHSA without additional stimulation with solutions containing glucose (Figure 3). In each case, the correct distribution of β-cells in the core of the aggregate was confirmed, while α-cells outside, creating a mantle surrounding β-cells. This distribution is correct and consistent with that obtained in our previous studies. In Figure 3, red color is less intense than green, which is associated with a greater ability to produce insulin than glucagon by aggregates incubated with PAHSA solutions. This is also confirmed by the observed differences in the levels of fluorescence intensity, which indicates a higher ability to secrete insulin and glucagon after stimulation with solutions with specific concentrations of PAHSA. The higher levels of insulin and glucagon fluorescence intensity for 60 µM and 100 µM of 5-PAHSA and a slight increase after incubation with 100 µM of 9-PAHSA were obtained. Moreover, the correct process of cell aggregation and the distribution of α-cells outside the aggregate and β-cells inside were confirmed (Figure 3).

### 3.3. The Effect of 5- and 9-PAHSA on Glucose Stimulated Insulin Secretion (GSIS)

Mice treated with PAHSA showed increased glucose tolerance and insulin sensitivity [19,20]. It was shown that 48-h incubation of isolated human pancreatic islets with PAHSA solution at a concentration of 40 μM increased GSIS. Moreover, the same study was performed for incubation only with palmitic acid (PA), and an increase in insulin secretion in response to glucose stimulation was not recorded [29]. However, it is not clearly defined what range of PAHSA concentration causes a direct negative or positive effect on β-cells. Different concentrations of PAHSA—usually a concentration of the order of nM or mM are often used in animal or isolated islets studies, frequently much higher to test the effect of PAHSA on GSIS and much lower to test its effect on the proliferation of pancreatic islet cells [22,23,29]. Glucose-stimulated insulin secretion (GSIS) is major parameter indicating functionality of pancreatic *β*-cells [30]. In the present study we checked whether the 48-h incubation with 5-PAHSA and 9-PAHSA affects GSIS. We performed ELISA test of insulin secretion after 48 h of incubation of pseudoislets with different concentrations (0 µM, 5 µM, 10 µM, 20 µM, 60 µM, 100 µM) of 5-PAHSA or 9-PAHSA, and cell stimulation with low (2.75 mM) or high (16.5 mM) concentrations of glucose. Thanks to the design two microchambers in one microfluidic system, cell stimulation of with low and high glucose solutions were performed simultaneously under the same environmental conditions. As was expected, at the lower concentrations of 5-PAHSA (5 µM, 10 µM, 20 µM), no significant increase in the level of insulin secretion was observed compared to the results obtained in the control (culture medium). On the other hand, after incubation with higher concentrations of 5-PAHSA, a significant increase in the level of insulin secretion was observed. The level of insulin secretion after incubation with 60 µM 5-PAHSA was 4,14 ng per pseudoislet after low glucose stimulation and 9.12 ng per pseudoislet after high glucose stimulation (a twofold increase compared with the control). After 48 h of incubation with 100 µM 5-PAHSA, the cells secreted 19.98 ng per islet at the stimulation with low glucose concentration solution, and 22.02 ng per islet in the high glucose environment (Figure 4). However, in this concentration (100 µM), no significant differences in the level of insulin secretion between stimulation with high and low glucose concentration were observed, which is not correct islet function. In opposite, no significant effect of 9-PAHSA at concentrations of 5 µM, 10 µM, 20 µM, 60 µM on GSIS was observed. Only after incubation with 100 µM 9-PASHA, an increase in insulin secretion was observed. The cells secrete 4.28 ng per islet and 10.71 ng per islet after stimulation with low and high glucose concentrations, respectively. However, it was more than two times lower increase in insulin secretion than after incubation with the same concentration of 5-PAHSA (Figure 4). It may also be associated with a lower rate of cell proliferation.

There are several literature reports confirming that HFD-induced insulin-resistant mice showed better glucose tolerance after treatment with 5-PAHSA [17,24]. Most of these studies are carried out on animal models. It has been proven that treatment with 5-PAHSA can increase glucose-stimulated (11 mM) insulin secretion by up to 30% [29]. Other reports deny positive effects of PAHSA on insulin secretion [18]. In our study, it was confirmed that PAHSA (especially 60 µM 5-PAHSA and 100 µM 9-PAHSA) might have a positive effect on increasing insulin secretion by pancreatic β-cells. Moreover, we examined insulin secretion after treatment with PAHSA isomers and after stimulation with both low and high glucose concentrations, in each case at least a 50% increase in insulin secretion relative to the control was confirmed (Figure 4). A similar trend of increasing insulin secretion was observed as in the studies on animal models presented in the literature, but a higher level of secretion was obtained after administration of lower concentrations of PAHSA, which had not been tested on pseudoislet model so far.

### 3.4. The Effect of 5- and 9-PAHSA on Glucagon Secretion

Glucagon plays an important role in regulation of blood glucose homeostasis. During hypoglycemia, insulin secretion is inhibited, glucagon is produced by α-cells, and when it reaches its maximum level, glucose is released from the liver. As the blood glucose level increases, the secretion of glucagon is inhibited, which causes an increase in insulin secretion [31,32]. Diabetes research usually focuses on β-cells, but it should be remembered that in order to maintain proper homeostasis of the blood glucose, the proper functioning of α-cells is equally important [33,34]. In the case of the studies on the effect of PAHSA on pancreatic cells, two parameters are usually analyzed: insulin secretion and the effect on body weight. There are no publications of in vitro or in vivo studies of an effect on glucose-stimulated glucagon secretion. There is one brief report compiled by Zhou et al. showed that in HFD-fed mice, treatment with PAHSA did not increase glucagon levels [20]. In our research, we incubated the pseudoislet model with various concentrations (5 µM, 10 µM, 20 µM, 60 µM, 100 µM) of 5-PAHSA and 9-PAHSA. After 48 h of incubation, the glucagon secretion from the pseudoislets in low (2.75 mM) and high (16.5 mM) of glucose was tested using the ELISA test (Figure 5). As expected, no statistically significant differences were noticed in glucagon secretion after incubation with lower concentrations of 5-PAHSA (5–60 µM). Only after 48 h of incubation with 100 µm 5-PAHSA, a twofold increase in glucagon secretion was noted after stimulation with a low glucose solution. Such an increase in glucagon secretion was not observed, for the same concentration of 5-PAHSA and stimulation with high glucose concentration. In the case of 9-PAHSA, it was noticed that glucagon secretion under low glucose conditions increased after incubation with 5 µM, 10 µM, 20 µM and 100 µM by, respectively: 0.39 ng, 0.09 ng, 0.36 ng, 0.23 ng per islet. However, in the case of stimulation with high glucose concentrations, an increase in glucagon secretion by 0.30 ng per islet was observed only after incubation with 60 µM 9-PAHSA. There was no such marked increase in secretion under both high and low glucose conditions as in the case of insulin secretion. The obtained results show that 5-PAHSA and 9-PAHSA in some concentrations may have a positive effect on glucagon secretion from α-cells, but practically only in conditions of low glucose concentration. Therefore, 5-PAHSA and 9-PAHSA are much better suited for administration of impaired insulin secretion in disease states but may be ineffective in increasing/reducing glucagon secretion.

## 4. Conclusions

The aim of this study was to determine the possibility of applying the developed model to the study of therapeutic agents (direct influence of two PAHSA isomers) and to determine the compatibility of cell responses in vivo and in vitro. Two aspects were the most important here: proof that PAHSA has a chance of being a completely safe therapeutic agent for type 2 diabetes and proof that our Lab-on-a-chip system can be used as a universal device for research on potential therapeutic agents and observation multiple responses (from single cell structures) at the same time. We decided to choose two isomers of PAHSA: 5-PAHSA because this form is the most severely reduced in all adipose tissue in mice and human resistant to insulin and 9-PAHSA which is the most abundant isomer in adipose tissue in mice and humans. There are several literature reports on the above-mentioned compounds, and the argument in their favor is that these substances can be completely safe due to their natural occurrence in the body [18]. Research on these compounds is based mainly on the introduction of PAHSA into the rodent’s diet and long-term determination of insulin levels and weight gain or the study of insulin secretion on isolated islets. Contrary to this, we present research using 3D pancreatic islet model (pseudoislet) under flow conditions. In this study, we determined the effect of a wide range of 5-PAHSA and 9-PAHSA concentrations on cell proliferation, change in islet mass and their unique structure. Moreover, we investigated the activity of pseudoislets, insulin and glucagon profile intensity, and the effect of PAHSA treatment on glucose stimulated insulin secretion and on glucose stimulated glucagon secretion. Due to developing the proper geometry and environment in developed model we were able to better mimic in vivo conditions, reduce the time needed to obtain a functional pseudoislet model and test multiple compound concentrations at the same time. In this study, we have proved that incubation with 5-PAHSA allowed for even a twofold increase in cell proliferation in the following days of culture, and thus increased their mass and higher insulin secretion levels. On the other hand, the degree of cell proliferation, their mass, and insulin secretion increased after incubation with 9-PAHSA but was lower than after incubation with 5-PAHSA in the same concentrations. Moreover, there was an increase in glucagon secretion after treatment with PASHA isomers, which was higher for 5 µM, 20 µM and 100 µM 9-PAHSA than for the same 5-PAHSA concentrations. In our research, we report that PAHSA isomers may have potential therapeutic properties and examined their direct effect on α- and β-cells. We believe that the best treatment results will be obtained with 60 µM 5-PAHSA and 100 µM 9-PAHSA. We also want to emphasize that the developed microsystem can be an ideal solution in screening tests, testing new drugs, or the impact of therapy on the development of diabetes, and thanks to its design it is possible to conduct research and observations in real time.

## Figures and Tables

**Figure 1 biosensors-12-00302-f001:**
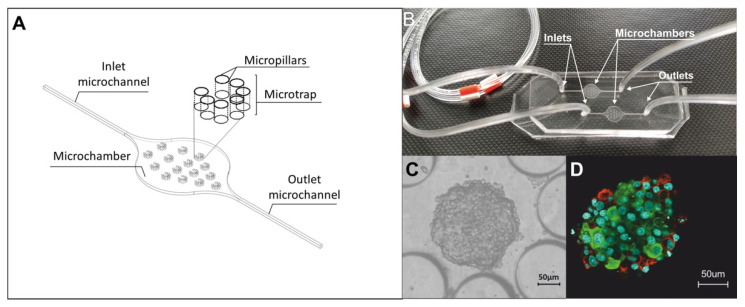
Islet-on-a-chip system. (**A**) Geometry of one microchamber of the islet-on-a-chip. (**B**) PDMS/PDMS microfluidic system. (**C**) INS-E and α-TC1-6 aggregates 24 h after cells introducing into islet-on-a-chip system. (**D**) Three-dimensional confocal image of the obtained aggregate. Confirmation of the localization of α- and β-cells by staining glucagon (conjugated with Alexa Fluor 594) (red cells) and insulin (conjugated with Alexa Fluor 488) (green cells). The cell nucleus is shown in blue (Hoechst staining).

**Figure 2 biosensors-12-00302-f002:**
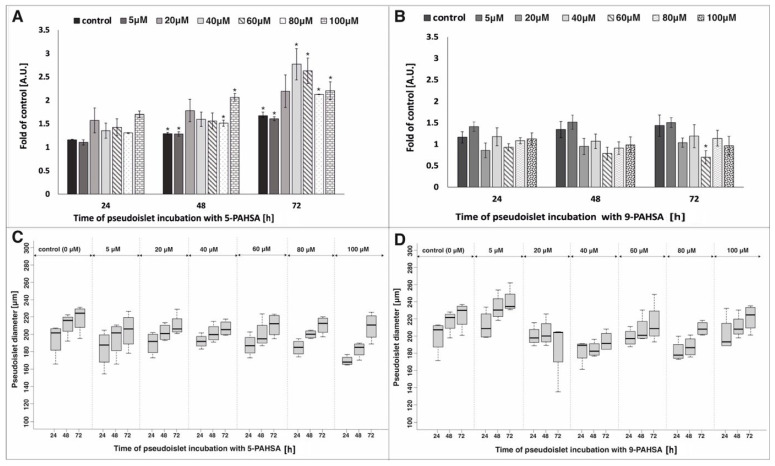
Proliferation and diameter of the pseudoislet after incubation with different concentration of 5-PAHSA (**A**,**C**) and 9-PAHSA (**B**,**D**). The results presented in the graphs were related to the measurement of the degree of proliferation 24 h after introducing the cells into Lab-on-a-chip system (without incubation with 5-PAHSA and 9-PAHSA), n = 3 * *p* < 0.05.

**Figure 3 biosensors-12-00302-f003:**
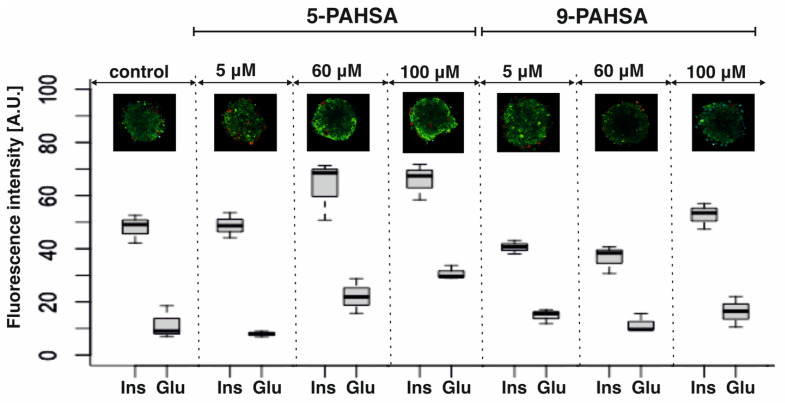
Immunostaining process of the pseudoislet. Confirmation of the localization of α- and β-cells by staining glucagon (conjugated with Alexa Fluor 594, red cells) and insulin (conjugated with Alexa Fluor 488, green cells). The analysis of the intensity of individual hormones was performed based on images obtained in a confocal microscope with the use of the CellSens Dimension program, n = 3.

**Figure 4 biosensors-12-00302-f004:**
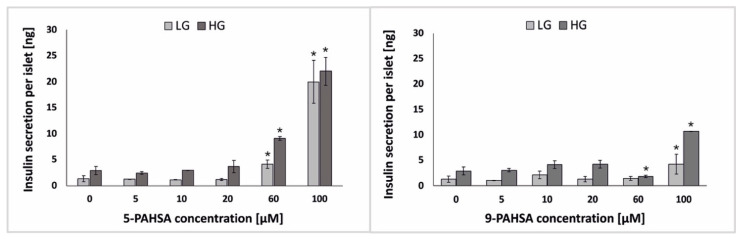
The level of insulin secretion after pseudoislet treatment with 5-PAHSA (**left**) and 9-PAHSA (**right**) and stimulation with low glucose (LG; 2.75 mM) and a high glucose (HG; 16.5 mM) solution. n ≥ 3. * *p* < 0.05.

**Figure 5 biosensors-12-00302-f005:**
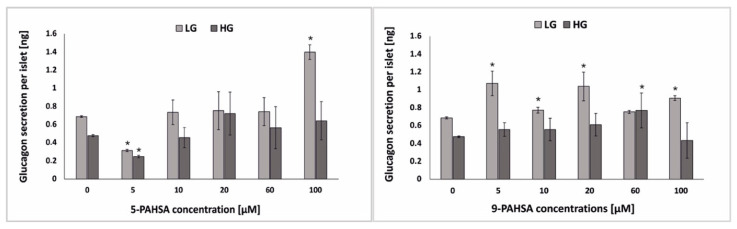
The level of glucagon secretion after pseudoislet treatment with 5-PAHSA (**left**) and 9-PAHSA (**right**) and stimulation with low glucose (LG; 2.75 mM) and a high glucose (HG; 16.5 mM) solution. n ≥ 3. * *p* < 0.05.

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
