# Peer review of "Investigation of the Therapeutic Potential of New Antidiabetic Compounds Using Islet-on-a-Chip Microfluidic Model"

_biosensors, 2022, doi:10.3390/bios12050302_

Round 1
Reviewer 1 Report
Summary
This paper presents the use of an islet-on-a-chip model to study the effects of 5-PAHSA and 9-PAHSA on cell proliferation, morphology, insulin and glucagon expression, and glucose-stimulated insulin and glucagon secretion. It demonstrates how the islet-on-a-chip model can be used to quickly assess certain effects of drugs on the pancreatic islet in an environment similar to in vivo. The paper’s results are presented clearly and in a logical order, the methodology is sound, and the conclusions are generally backed by the data, but some additions to the discussion are suggested.
Specific Comments
- Lines 88, 355-56: The authors claim their islet-on-a-chip model can be used as a universal screening device for new drugs in the introduction and in the conclusion. I think this claim is too broad given the scope of the paper and its main aims. The use of the islet-on-a-chip model as a universal screening tool is not investigated in this paper. This would entail, at a minimum, a more detailed analysis tying the islet-on-a-chip model results to similar experiments in vivo. Although this is certainly a step in that direction, I think the authors should narrow their claim.
- Lines 94-114: This section describes the construction of the islet-on-a-chip model. While this section describes the construction of the chip in enough detail and appropriately refers to the earlier work, it would greatly benefit the reader to include in the present work a simple illustration of the chip.
- Lines 214-220, 232: The authors claim the diameters of the aggregates increase after treatment with 5- and 9-PAHSA. This claim is backed up by Figure 1. However, it seems noteworthy that the diameters appear to be less than the control. I wonder what the authors see as the implication of this data.
- Lines 240-241, 256-258: The authors maintain the morphology of the pseudoislet is very important. However, they do not discuss whether their pseudoislets match the predicted distribution of 60% beta cells in the core and 25% alpha cells in the peripheral, but instead jump straight to the conclusion in lines 256-258. Furthermore, lines 256-258 should precede the discussion in lines 252-256 as it relates to the morphology and not to the fluorescence intensities of insulin and glucagon.
- Line 247- Aligning all the optical images with each other would improve the presentation of this figure.
Author Response
Warsaw, 02.05.2022
Dear Reviewer,
Thank you for your time evaluating our manuscript. Please find enclosed our revised manuscript (ID: biosensors-1703380) “Investigation of the therapeutic potential of new antidiabetic compounds using Islet-on-a-chip microfluidic model” According to the comments, we have introduced all needed explanations and corrections in the manuscript.
We hope that you find the changes to our manuscript satisfactory and consider it for publication.
Sincerely yours,
Patrycja Sokołowska
Reviewer 1:
Summary
This paper presents the use of an islet-on-a-chip model to study the effects of 5-PAHSA and 9-PAHSA on cell proliferation, morphology, insulin and glucagon expression, and glucose-stimulated insulin and glucagon secretion. It demonstrates how the islet-on-a-chip model can be used to quickly assess certain effects of drugs on the pancreatic islet in an environment similar to in vivo. The paper’s results are presented clearly and in a logical order, the methodology is sound, and the conclusions are generally backed by the data, but some additions to the discussion are suggested.
Specific Comments
- Lines 88, 355-56: The authors claim their islet-on-a-chip model can be used as a universal screening device for new drugs in the introduction and in the conclusion. I think this claim is too broad given the scope of the paper and its main aims. The use of the islet-on-a-chip model as a universal screening tool is not investigated in this paper. This would entail, at a minimum, a more detailed analysis tying the islet-on-a-chip model results to similar experiments in vivo. Although this is certainly a step in that direction, I think the authors should narrow their claim.
We agree with the reviewer. The used statement that "islet-on-a-chip model can be used as a universal screening device for new drugs" should be confirmed by additional research, but it is undoubtedly a demonstration of the usefulness of our device in this direction. According to the reviewer suggestion, we have narrowed our claim.
- Lines 94-114: This section describes the construction of the islet-on-a-chip model. While this section describes the construction of the chip in enough detail and appropriately refers to the earlier work, it would greatly benefit the reader to include in the present work a simple illustration of the chip.
According to the reviewer suggestion an illustration of the islet-on-chip system and the obtained cell structures after 24 hours of culture have been included.
- Lines 214-220, 232: The authors claim the diameters of the aggregates increase after treatment with 5- and 9-PAHSA. This claim is backed up by Figure 1. However, it seems noteworthy that the diameters appear to be less than the control. I wonder what the authors see as the implication of this data.
In our device, in each of the 15 traps, we usually obtain aggregates with diameters in the range of 185-200 μm. Indeed, the aggregates used for stimulation with some PAHSA concentrations were slightly smaller than those used for the tests under control conditions. Nevertheless, in this study we wanted to emphasize how the aggregate diameter changes during the entire time of the culture (PAHSA stimulation). In this case, we observe a greater increase in the diameter of aggregates stimulated with higher concentrations of PAHSA in the following days of culture. It is very clearly noticeable in the case of 100 μM 5-PAHSA, where the aggregate diameter increases from 175 μm (24h of the culture) to 220 μm (72h culture) which results an increase of 35 μm during the time of the culture. On the other hand, under control conditions, we observe an increase from 210 μm to 230 μm, which results a diameter increase of 20 µm.
- Lines 240-241, 256-258: The authors maintain the morphology of the pseudoislet is very important. However, they do not discuss whether their pseudoislets match the predicted distribution of 60% beta cells in the core and 25% alpha cells in the peripheral, but instead jump straight to the conclusion in lines 256-258. Furthermore, lines 256-258 should precede the discussion in lines 252-256 as it relates to the morphology and not to the fluorescence intensities of insulin and glucagon.
Thank you for this reviewer's comment, we have indeed omitted the conclusion that was important for the reader regarding the composition and location of the cells. According to the reviewer suggestion necessary clarifications have been added.
- Line 247- Aligning all the optical images with each other would improve the presentation of this figure.
According to the reviewer suggestion figure 3 has been corrected.

Reviewer 2 Report
This paper is an excellent application of Islet-on-a-chip device previously developed in authors' laboratory, to demonstrate the usefulness of this device to select potential therapeutics for diabetes mellitus treatment.
The authors present the data of cell immunostaining and ELISA results of insulin and Glucagon secretion when the solution of drug candidates, 5-PAHSA and 9-PAHSA are loaded into this device. The comparisons with animal study papers validate the feasibility of using this device.
Prior to the publication of this paper, the following suggestions should be addressed by the authors.
(1) The rationale of using 10 micro-L/min as the infusion flow rate should be provided. Cell deformation has to be considered to make sure the secretion is only determined by chemical simulation.
(2) Professional English proofreading is needed for the revision.
Author Response
Warsaw, 02.05.2022
Dear Reviewer,
Thank you for your time evaluating our manuscript. Please find enclosed our revised manuscript (ID: biosensors-1703380) “Investigation of the therapeutic potential of new antidiabetic compounds using Islet-on-a-chip microfluidic model” According to the comments, we have introduced all needed explanations and corrections in the manuscript.
We hope that you find the changes to our manuscript satisfactory and consider it for publication.
Sincerely yours,
Patrycja Sokołowska
Reviewer 2:
This paper is an excellent application of Islet-on-a-chip device previously developed in authors' laboratory, to demonstrate the usefulness of this device to select potential therapeutics for diabetes mellitus treatment. The authors present the data of cell immunostaining and ELISA results of insulin and Glucagon secretion when the solution of drug candidates, 5-PAHSA and 9-PAHSA are loaded into this device. The comparisons with animal study papers validate the feasibility of using this device.
Prior to the publication of this paper, the following suggestions should be addressed by the authors.
- The rationale of using 10 micro-L/min as the infusion flow rate should be provided. Cell deformation has to be considered to make sure the secretion is only determined by chemical simulation.
Thank you for the reviewer's comment. The parameters for the introduction of the cells suspension and the administration of all solutions were selected on the basis of simulation (performed using Microelectromechanical Systems (MEMS) simulation module of COM- SOL Multiphysics software) and confirmed experimentally with the use of proliferation/viability tests, immunofluorescence staining and microscopic observations. We presented these results in our previous publication (Sokolowska, P.; Zukowski, K.; Janikiewicz, J.; Jastrzebska, E.; Dobrzyn, A.; Brzozka, Z. Islet-on-a-Chip: Biomimetic Micropillar-Based Microfluidic System for Three-Dimensional Pancreatic Islet Cell Culture. Biosens Bioelectron 2021, 183, 113215, doi:10.1016/j.bios.2021.113215.). In addition, all our tests have been performed for control as well (culture medium only), so by observing the differences we can be sure that the secretion is determined only by chemical simulation. To make this part more understandable to the reader necessary clarifications have been added.
- Professional English proofreading is needed for the revision.
According to the suggestion of the reviewer manuscript was checked by native English speaker and the language corrections was made.
